# Dynamics of Polymer Membrane Swelling in Aqueous Suspension of Amino-Acids with Different Isotopic Composition; Photoluminescence Spectroscopy Experiments

**DOI:** 10.3390/polym13162635

**Published:** 2021-08-07

**Authors:** Nikolai F. Bunkin, Polina N. Bolotskova, Elena V. Bondarchuk, Valery G. Gryaznov, Valeriy A. Kozlov, Maria A. Okuneva, Oleg V. Ovchinnikov, Nikita V. Penkov, Oleg P. Smoliy, Igor F. Turkanov

**Affiliations:** 1Department of Fundamental Sciences, Bauman Moscow State Technical University, 2-nd Baumanskaya Str. 5, Moscow 105005, Russia; bolotskova@inbox.ru (P.N.B.); v.kozlov@hotmail.com (V.A.K.); neonlight0097@gmail.com (M.A.O.); 2Prokhorov General Physics Institute of the Russian Academy of Sciences, Vavilova Str. 38, Moscow 119991, Russia; 3Concern GRANIT, Gogolevsky blvd., 31, bldg. 2, Moscow 119019, Russia; info@npo-qt.ru (E.V.B.); gryaznov.v@granit-concern.ru (V.G.G.); office@granit-concern.ru (O.V.O.); smoliy.o@granit-concern.ru (O.P.S.); turkanov.i@granit-concern.ru (I.F.T.); 4Federal Research Center “Pushchino Scientific Center for Biological Research of the Russian Academy of Sciences”, Institute of Cell Biophysics of the Russian Academy of Sciences, Institutskaya Str. 3, Pushchino 142290, Russia; nvpenkov@rambler.ru

**Keywords:** polymer membranes, photoluminescence spectroscopy, quenching of luminescence, amino-acids, polymer swelling, dynamic light scattering, isotopic effects, ζ-potential, logic gates, binary cells

## Abstract

In photoluminescence spectroscopy experiments, the interaction mode of the polymer membrane Nafion with various amino-acids was studied. The experiments were performed with physiological NaCl solutions prepared in an ordinary water (the deuterium content is 157 ± 1 ppm) and also in deuterium-depleted water (the deuterium content is ≤1 ppm). These studies were motivated by the fact that when Nafion swells in ordinary water, the polymer fibers are effectively “unwound” into the liquid bulk, while in the case of deuterium-depleted water, the unwinding effect is missing. In addition, polymer fibers, unwound into the liquid bulk, are similar to the extracellular matrix (glycocalyx) on the cell membrane surface. It is of interest to clarify the role of unwound fibers in the interaction of amino-acids with the polymer membrane surface. It turned out that the interaction of amino-acids with the membrane surface gives rise to the effects of quenching luminescence from the luminescence centers. We first observed various dynamic regimes arising upon swelling the Nafion membrane in amino-acid suspension with various isotopic content, including triggering effects, which is similar to the processes in the logical gates of computers.

## 1. Introduction

Nafion™ (for more details see review [1]) is an ionic copolymer consisting of a perfluorocarbon backbone (similar to Teflon) and cross-linked terminal sulfonate groups. Nafion has high chemical and thermal stability and is being actively studied as a matrix for hydrogen fuel cells. In this case, sulfonate groups have hydrophilic properties, while perfluorocarbon chains are hydrophobic ones. In anhydrous Nafion, hydrophilic sulfonate groups HSO_3_ are most often collected in rounded or rod-shaped reverse micelles immersed in a hydrophobic matrix. When swollen in water, Nafion exhibits amphiphilic properties: various structures, consisting of hydrophilic and hydrophobic segments, are formed. These structures are highly sensitive to changing the concentration of water inside the polymer matrix, which leads to the formation of cylindrical micelles in the form of pass-through cylindrical channels. Due to the dissociation of sulfonate groups, occurring upon swelling inside these channels, the polymer matrix allows protons to penetrate into these channels (which is used in hydrogen energetics), but does not allow anions and non-polar molecules such as O_2_ to pass through these channels. We note that the effect of moisture is critical not only for polymer matrices, but also for crystalline multilayer meso-porous films, see, for example, the recent work [2]. At the Nafion-water interface, dissociation of terminal sulfonate groups develops according to the reaction R—SO_3_H + H_2_O ⇔ R—SO_3_^−^ + H_3_O^+^, i.e., the proton moves to the water bulk, and the surface of Nafion in water exhibits acidic properties. In [3], the distribution of the pH value across water layer close to the Nafion interface was measured as a function of the distance from the membrane surface; it was found that within 1 mm from the membrane surface, pH ≈ 3. In addition, as was shown in [4], the polymeric chains micelles are oriented in parallel to the membrane surface if the membrane is adjacent to a gas medium, while if the membrane is adjacent to water, micelles are oriented perpendicular to the membrane surface. Upon swelling in water, the volume of Nafion hydrophilic phase increases, and reverse micelles are connected to each other forming a three-dimensional network [5]. The volume fraction of the hydrophobic phase decreases, and this phase decomposes into individual micelles. Nafion properties depend on various impurities in the system. For example, it was shown in [6] that additions of H_2_S of about 10 ppb to pure hydrogen resulted in a significant decrease in the proton conductivity through a polymer membrane in a hydrogen fuel cell, i.e., to a decrease in the fuel cell efficiency.

When Nafion swells in water, an area of a special structure close to the membrane surface is formed. It turned out that micron-sized colloidal particles are effectively pushed out from this area; this is why it was termed the “exclusion zone”, see monography [7] and references therein. The exclusion zone is about 200 microns thick; this result was obtained in optical microscopy experiments. At the same time, it is reckoned that a solid surface can modify an adjacent water layer only within 1–2 nm; this size corresponds to the radius of the so-called dispersion forces, see, for instance, [8]. The most relevant information on studies of the exclusion zone, including a detailed discussion on the physical mechanisms of formation of this structure, can be found in the review [9].

As was shown in our works [10,11,12], the exclusion zone formation is associated with the effect of “unwinding” of polymer fibers from the membrane surface into the water bulk. Since the unwound fibers do not completely detach from the membrane surface, these fibers resemble glycocalyx (extracellular matrix)-polysaccharide fibers on the lipid bilayer of cell membrane (see, for example, [13]). In this case, a topological structure of the “hard brush” type is formed; it was shown in [10] that near the membrane surface this structure is similar to a colloidal crystal formed by negatively charged rod-like polymer particles. Additionally, in [11,12] it was shown that the area occupied by the unwound polymer depends on the deuterium content in water. In fact, for ordinary water (the deuterium content is 157 ± 1 ppm, see [14]), the size of this area is about 200 μm, which is close to the exclusion zone size. At the same time, for deuterium depleted water (DDW, the deuterium content is ≤1 ppm), the effect of unwinding is missing. In [12], it was hypothesized that Nafion membrane is similar to a cell membrane in the sense that both the cell membrane and Nafion plate in water are surrounded by a “brush” made of polymer fibers. Therefore, there is of interest to study the interaction of various amino-acids with Nafion, taking into account the unwinding effect.

In this regard, it should be noted that the proteins interaction with the Nafion membrane was studied mainly in the context of encapsulation of redox-active proteins, such as glucose oxidase, cytochromes, ferredoxin, and the complex of the cyanobacterial photosystem, see Ref. [15]; properties of the photosystem based on Nafion are described in details in review [16]. It was shown that, upon contact of the Nafion surface with a protein suspension, protein macromolecules do not penetrate into the polymer matrix. To obtain matrix-encapsulated proteins, protein films and a polymer membrane are usually placed in conductive cells. In this case, very thin (up to 1 μm) Nafion films are employed, since the polymer conductivity decreases with increasing the film thickness.

A large number of papers are devoted to the encapsulation of glucose oxidase inside Nafion; these studies open up prospects for highly sensitive bioelectric glucose sensors, including implantable ones, which are important for diabetic patients, see, e.g., [17]. The authors of this study measured activity of positively charged cytochrome and neutral cytochrome b5 and cytochrome c551 encapsulated in the Nafion membrane; it was found that neutral proteins remain redox-active, while positively charged cytochrome loses redox activity. The Nafion interaction with human blood, tissue fluid redox-passive proteins, such as albumin, fibrinogen, fibronectin, and undiluted blood serum was studied in [18,19].

The interaction of various bacteria with the Nafion surface, taking into account an exclusion zone formation, was also investigated. In this regard, it is reasonable to note the recent work [20], where in the experiment based on fluorimetry it was shown that various bacteria in an ordinary water suspension are effectively repelled from the Nafion surface at a distance of about the exclusion zone size.

An additional motivation of our research is the fact that, as far as we know, a logical binary cell was implemented on the basis of amino-acids. In a recent was [21] a system that includes an organometallic crystal matrix, into which trivalent terbium ions, divalent copper ions and aspartic acid are introduced, was described. It turned out that trivalent terbium ions are an effective fluorophore, while bivalent copper ions serve as a luminescence quencher for this fluorophore. At the same time, aspartic acid forms a complex with copper ion, which neutralizes the luminescence quenching function of this ion, i.e., the luminescence of Tb^3+^ is efficiently recovered. This work shows that a high luminescence level (for example, in the absence of Cu^2+^-ion, or in the presence of both Cu^2+^-ion and aspartic acid) can be conventionally considered as “1” of a binary logic cell, while a low luminescence level in the presence of copper ions without aspartic acid can be conventionally considered as “0” of the binary cell. Since our experimental technique is based on the excitation of luminescence from the Nafion surface (see below), we are interested in the effects of quenching/recovery of luminescence by the amino-acids in a system with Nafion, i.e., the possibility of realizing a logical binary cell on the basis of a Nafion membrane immersed in a suspension of amino-acids.

To summarize, we are not aware of any works in which the specifics of the interaction of amino-acids with the Nafion membrane surface under the conditions of unwinding of polymer fibers had been studied. It is of special interest to investigate this interaction in the absence of the unwinding effect (in this case, the amino-acid suspensions should be prepared on the basis of DDW). Concluding this section, we should remind that an acidic environment with pH ~3 exists in water near the Nafion surface; this should be taken into account in experiments.

## 2. Materials and Methods

To carry out experiments with photoluminescence spectroscopy of Nafion in amino-acid suspensions, an experimental setup, illustrated in Figure 1 (see [10,22]), was designed. 

The experimental setup includes a continuous wave diode laser (1) with a wavelength λ = 369 nm, a spectrometer (6) with a spectral range of 240–1000 nm and 2 nm-resolution, a multi-mode optical fiber (2) with the 50 μm- core diameter transferring the laser radiation (1) into the cell (3) with a liquid sample, and a similar optical fiber (5), through which the luminescence signal was fed the spectrometer input. To avoid sufficiently intense laser radiation from entering the spectrometer (6) input. The fiber (5) was equipped with a refocuser and a light filter that cut-off radiation within the range λ < 350 nm (not shown in Figure 1). Spectra were processed by a personal computer (7). The cylindrical cell (3) was made of Teflon and had a radius of 2 cm. The Nafion plate (4) was fixed on a stage (8) with a micrometric horizontal feed. This feed was used for precise alignment of the Nafion plate and the optical axis; the optical axis coincided with geometric axis of the cylindrical cell, i.e., the Nafion plate was illuminated by laser light in the grazing incidence geometry. We used Nafion N117 plates (Sigma Aldrich, St. Louis, MO, USA) with a thickness of *L*_0_ = 175 μm and an area of 1 × 1 cm^2^.

Laser diode radiation stimulated luminescence from the Nafion plate in the spectral range 400–600 nm. In our previous work [11] it was shown that the luminescence centers are terminal sulfonate groups. As shown in [4], when the Nafion plate is immersed in water, the surface bundles of polymer fibers tend to be oriented perpendicular to the surface, i.e., the terminal sulfonate groups are located mainly at the polymer–water interface. This is why the luminescence in our experiments was excited in the geometry of grazing incidence of the pump radiation.

Since Nafion is transparent to radiation in the visible range, the luminescence radiation, reflected from the cylindrical wall of the cell, was concentrated along the optical axis (5), and then entered the spectrometer input (6). In the absence of Nafion, luminescence in this spectral range had not been detected in saline solutions prepared in natural water/DDW, as well as in amino-acid suspensions for both solutions. This had been purposefully verified in the experiments.

Before performing experiment, the luminescence spectrum of dry Nafion was measured; while measuring this spectrum, the cell was not filled with a liquid sample. Using a micrometer screw of the stage (8), the Nafion plate was oriented in such a way that the luminescence signal at the spectral maximum (460 nm) reached its maximum value; this corresponds to the optimum orientation of the plate along the optical axis. Then, the test liquid was poured into the cell; the moment of filling the cell corresponds to the starting point of the Nafion swelling time.

The studied amino-acids were glycine, alanine, histidine, lysine, as well as aspartic and glutamic acids, purchased from Sigma Aldrich, St. Louis, MO, USA. Reagent-Grade NaCl (Sigma Aldrich, St. Louis, MO, USA) was used to prepare saline solutions. Deionized water (deuterium content 157 ± 1 ppm) with a resistivity of 18 MΩ × cm at 25 °C, refined by a Milli-Q apparatus, was employed as a solvent. Also for the preparation of saline NaCl solutions, DDW (deuterium content ≤ 1 ppm), purchased from Sigma Aldrich, St. Louis, MO, USA, was employed.

To find the sizes of the amino-acids under study and the values of their zeta-potential, dynamic light scattering (DLS) experiments were carried out with a Zetasizer Nano ZS system (Malvern, UK) equipped with a continuous wave He−Ne laser at a wavelength of λ = 633 nm (maximum intensity is 4 mW) and a temperature controller; the scattering angle was 173° (the dynamic light scattering technique was comprehensively described in monographs [23,24]). In our experiments, hydrochloric acid (Sigma Aldrich, St. Louis, MO, USA) was added to an amino-acid suspension to reach the value pH = 3. As already noted, this was necessary to create conditions close to those realized at the interface of Nafion in water. In order to exclude the contribution from aggregates of amino-acids and gas nanobubbles, the suspensions were prefiltered with a 200 nm-pore filter manufactured by TPP (Trasadingen, Switzerland); such a technique has been applied in our recent experiments [25,26].

We also investigated photoluminescence from a colloidal aqueous suspension of Nafion particles. For this purpose, a 175 μm-thick Nafion sheet was ground into colloidal particles powder with the particle average size of 12 μm (this size was measured using an optical microscope) with a colloid mill (manufactured by Komandarm LLC, Novosibirsk, Russia).

Concluding this section, it seems meaningful to give some comments on the luminescence signal, which we measured. Since water molecules penetrate into the polymer matrix during the Nafion swelling, the surface density of sulfonate groups should decrease. The intensity *I*(*t*) of the luminescence signal in the spectral maximum (λ = 460 nm), where *t* is the swelling time, is given by the formula (see, e.g., [10])
*I*(*t*) = *A*_0_ + *k I*_pump_ σ_lum_*n*_Naf_(*t*) *V*,(1)
where *I*_pump_ is the pump intensity, *n*_Naf_ is the volume number density of luminescence centers (in our case, this is the density of sulfonate groups), *σ*_lum_ is the luminescence cross-section, *A*_0_~20–270 a.u. corresponds to the spectrometer spectral noise (this noise obviously includes a stray light), *k* is a dimension coefficient, which is controlled by the spectrometer sensitivity, *V* is the luminescence volume. Obviously, under the condition σ_lum_ = const (there are no effects of quenching/enhancement of luminescence upon irradiation of sulfonate groups by the pump radiation), the luminescence intensity *I*(*t*) is completely determined by the density *n*_Naf_(*t*), which exponentially decreases upon swelling (see our recent work [22]). In this case, the experimental curves *I*(*t*) can be approximated by exponential functions.

The experimental dependences presented below correspond to averaging over 5 successive measurements; the bars in figures show the confidence intervals.

## 3. Experimental Results

Figure 2 shows the dependences of the luminescence intensity *I*(*t*) in the spectral maximum vs. the soaking time in ordinary water and DDW in the absence of NaCl and amino-acids; it is seen that the dependences are well approximated by decaying exponential functions, and the free constants, pre-exponential factors, and relaxation times are similar for both liquids. Figure 3 exhibits the dependences of the luminescence intensity *I*(*t*) in the spectral maximum vs. the soaking time in ordinary water-based and DDW-based physiological solutions (0.9 vol.% NaCl) without amino-acids. As in the graphs of Figure 2, the experimental dependences are well approximated by decaying exponential functions, and the free constants, pre-exponential factors, and relaxation times are very close for both liquids.

The amino-acids under study were characterized in DLS experiments at pH = 3. The weight concentration for all amino acids in these experiments was 20 g/L; such a high concentration of amino-acids in this experiment is due to the very small scattering cross section of particles, the size of which is on the order of several Angstroms. For each amino-acid suspension, the dynamic viscosity coefficient was previously measured. For each amino-acid, 5 consecutive measurements were made; the mean values of ζ-potentials and hydrodynamic diameters were found. Note that all liquid samples were pre-filtered on a 200 nm-porous membrane to suppress amino-acid aggregation and exclude the contribution from a nanobubble phase (according to [25,26], the diameter of gas nanobubbles is ~300 nm, i.e., such particles should be removed by filtration). The hydrodynamic radius *R* and the ζ-potential *φ* obey the formula:
(2)$$\varphi =\frac{q}{4\pi \varepsilon_{0}\varepsilon R}$$
where *q* is the value of charge, *ε* is the dielectric constant of water, *ε* = 80, (1/4*πε*_0_) = 9 × 10^−9^ m/F. From Equation (2) the charge *q* was calculated for each amino-acid. In DLS experiments, we were unable to measure the values of *R* and *φ* for the glycine suspension. In Figure 4a,b we exhibit the histograms of DLS intensity vs. hydrodynamic diameters of the scatterers; in order not to overload the figure, we present data only for lysine (panel (a)) and glutamic acid (panel (b)).

Table 1 shows the results of dynamic light scattering experiment; the literature data on the sizes and the pH values at isoelectric point (pI) of the studied amino-acids are also given.

As follows from the data of Table 1, the charge value for all amino-acids is less than the electron charge (*e* = −1.6 × 10^−19^ Q), which is evidently due to the screening by ions, dissolved in liquid. Note that, in accordance with the data [27], the isoelectric points for glutamic and aspartic acids correspond to pI = 3.2 and 3.0 respectively, while for lysine the isoelectric point is pI = 9.8. Therefore, the glutamic and aspartic acids at pH = 3 are indeed near the isoelectric point, while lysine is far from the isoelectric point and should be positively charged. Thus our experimental results basically do not contradict the literature data. The contradictions to the data [27] were obtained for histidine and alanine: according to [27], the isoelectric points for histidine and alanine are pI = 7.6 and 6.01 respectively, i.e., at pH = 3 these amino-acids should have a positive charge, while we obtained that they are slightly negative. Apparently, for a more accurate measurement of ζ-potential for the Angstroms-sized particles, more precise experiments should be performed; to the best of our knowledge, no one has yet determined the values of amino-acids charge by measuring the zeta-potential with Zetasizer Nano ZS system. Basically, to measure the sign of the charge on an amino-acid at a fixed pH value, very different experimental techniques are used, see [27]. 

Prior to performing experiments with amino-acids suspensions, it was necessary to be convinced that, firstly, the photoluminescence from the amino acids is not stimulated in our experiments. For this purpose, amino-acids suspensions with a weight concentration of 1 g/L, prepared in ordinary water-based saline solution (0.9 vol.% NaCl) at pH = 5.5 (in these experiments, HCl acid was not added to liquid samples) were poured into the cell (3) in Figure 1; Nafion plate was not used in these experiments. It turned out that the luminescence of amino-acids is not excited by laser irradiation at the wavelength of λ = 369 nm; for more detailed information on the spectral characteristics of the amino-acid photoluminescence, see monograph [32]. Besides, it was necessary to make sure that there exist (or, on the contrary, not exist) effects of quenching/enhancement of luminescence by amino-acids (the mechanisms of luminescence quenching are described in [33]). Indeed, if the luminescence cross-section *σ*_lum_ is not constant, the *I*(*t*) dependence will be more complex than that, shown in Figure 2 and Figure 3. The Nafion plate was first ground into powder with a colloid mill; the mean size of the Nafion particles was ~12 μm (this was controlled with an optical microscope). Then the colloidal particles were dissolved in ordinary water-based 0.9 vol.% NaCl solution with pH = 3 (this value was attained by adding HCl acid, Sigma Aldrich, St. Louis, MO, USA). As follows from the graphs in Figure 5, before adding amino acids to the Nafion particles solution, the luminescence level was approximately the same and was controlled by the Nafion particles concentration (to improve the quality of the presentation, we used solutions with slightly different concentrations of Nafion particles). Then at the 20-th minute an amino-acid suspension of 1 g/L concentration, prepared in ordinary water-based 0.9 vol.% NaCl solution, was added to the colloidal mixture of Nafion particles; the volumes of the colloidal mixture and the amino-acid suspension were the same. This resulted in a sharp decrease in the luminescence, i.e., we are dealing with quenching. The effect of luminescence quenching was observed for all studied amino-acids, (in order not to overburden the Figure 5 we present the dependences only for aspartic acid, glycine and lysine). As is known, luminescence quenching is a very fast process, see ref. [33]. In our case the quenching develops within 1–2 min, which was due to the processes of mixing of amino-acids and the Nafion particles. As follows from the dependencies in Figure 5, after adding the amino-acid suspension and the completion of the mixing processes, the luminescence level reaches once again a stationary level. Thus, the duration of this experiment was not crucial; the experiment was lasted 40 min.

In Figure 6a–f we exhibit the dependences of *I*(*t*) for suspensions of various amino-acids, prepared in ordinary water-based and DDW-based saline solutions of 0.9 vol.% NaCl. The *I*(*t*) dependence is well approximated by two exponential functions, which indicates the presence of at least two dynamic modes. 

Let us first consider the dependences, which are related to the suspensions, based upon ordinary water (the effect of unwinding the polymer fibers is present). According to our model, the luminescence is quenched first, which is accompanied by a sharp decrease in the luminescence intensity (the characteristic times of luminescence quenching are very short, see [33]). The quenching is followed by the slower process, which is associated with the polymer fibers unwinding and swelling of Nafion. 

Table 2 shows the values of the rates (in a.u.) of decaying the luminescence intensity in *e* = 2.781828 times (the so-termed Euler number) for the first (*v*_1_) and second (*v*_2_) exponentials, as well as the values *A* of free constants (in a.u.) for all amino-acids in ordinary water-based saline solution (0.9 vol% NaCl). Without loss of generality, in the case of a suspension based on DDW, the *I*(*t*) dependence is also approximated by two decaying exponential functions. The rates *v*_1_ and *v*_2_ and values *A* of the free constants for suspensions prepared in DDW-based saline 0.9 vol.% NaCl are collected in Table 3. It should be noted that the estimates of *v*_1_ are very approximate, since our experimental setup is not designed to measure such high rates.

## 4. Discussion

In the absence of amino-acids, we are dealing with a single relaxation exponent only, see Figure 2 and Figure 3. For ordinary water/DDW, we have for the decay rates the of the exponents *v* = 47,723/(14.1 × 2.782) = 1217 a.u. and *v* = 44,234/(14.4 × 2.782) = 1104 a.u. correspondingly, see Figure 2. For ordinary water-based/DDW-based physiological solutions (0.9 vol.% NaCl) we have respectively *v* = 24,196/(12.73 × 2.782) = 683 a.u. and *v* = 23,022/(17.55 × 2.782) = 472 a.u. correspondingly, see Figure 3. As shown in [22], these rates are associated with the relaxation process of decreasing a surface density of sulfonate groups (luminescence centers) upon swelling. A decrease in the rate of swelling in a saline solution in comparison with pure water is due to the generation of nanobubbles on the membrane surface in the presence of ions, see [12]; these nanobubbles are effectively generated in electrolyte solutions (see [25,26]) and apparently prevent the membrane from swelling. 

As is seen from the Table 2 and Table 3, for suspensions of amino acids based on ordinary water, the ratio of *v*_1_/*v*_2_ ranges from 6584 (lysine) to 18 (histidine), while for suspensions of amino acids prepared on the basis of DDW, this ratio ranges from 643 (alanine) to 43 (glycine). Thus, the velocity ratios *v*_1_/*v*_2_ for suspensions based on ordinary water/DDW belong approximately to the same range. This is why the decay rates *v*_1_ and *v*_2_ are not the key parameter characterizing the differences between the behavior of amino-acids in ordinary water and DDW. 

For both types of amino-acid suspensions, the first exponent is associated with the luminescence quenching as the very fast process, see [33]. The second exponent corresponds to a slower relaxation process, which manifests itself after the completion of the luminescence quenching. For the suspensions prepared on the basis of DDW, free constants (luminescence intensities *I*(*t*) at *t* → ∞) belong to the range 83 < *A* < 1543 a.u., while for suspensions prepared on the basis of ordinary water, the free constants are within the range 5612 < *A* < 38,057 a.u.; recall that the initial (at *t* = 0) luminescence level corresponds to 65,000 a.u. We can say that the only qualitative difference between the behavior of the *I*(*t*) plots for suspensions of amino-acids prepared on the basis of ordinary water and DDW lies precisely in the values of the free constants. Thus it can be argued that the quenching process described by the first exponential is effectively interrupted for suspensions in ordinary water, while for suspensions based on DDW, the quenching process luminescence is completed (the level of luminescence decreases 40–800 times compared to the initial level). This can be explained by the effect of unwinding polymer fibers into the bulk of ordinary water-based saline solution; this effect is absent in the DDW-based saline solutions. Attraction forces, arising between the unwound polymer fibers in the water bulk, are due to an additional free energy: (3)$$G\left(a,r,T\right)\sim -\frac{\pi k_{B}T}{8rln^{2}\left(\frac{r}{a}\right)},$$
where *r* is the distance between parallel cylinders per unit length and *a* is the width of the cylinders, *k_B_* is Boltzmann constant, for more detail see [34]. This interaction appears due to correlations in charge fluctuations and leads to non-additive very long-range forces, wherein the attractive cooperative many-body interactions are opposed by the repulsive electrostatic double layer forces between the thin fibers. As was shown in [10], the combination of repulsive electrostatic forces and attraction forces leads to the formation of the ordered spatial structure of Nafion fibers close to the polymer surface. This structure can be considered as a special type of colloidal crystal, which was supported by the birefringence experiments, see [10].

We are unaware of scales *r* and *a*; however, we can use the data of Ref. [4]. In this work, in particular, the Nafion–water interface was studied with the atomic force microscopy, and it was found that polymer fibers in water near the membrane surface are predominantly oriented perpendicular to the surface, and the characteristic scale of the near-surface inhomogeneity is about 1.85 nm (see Figure 6b in [4]). This size should be close to the unwound polymer fibers diameter *a*. Assuming *r ≈ a*, the attractive force between polymer fibers will be very large due to the logarithmic term in the denominator. Further analysis requires taking into account the interaction between the unwound polymer fibers and amino acids located in the spaces between the fibers. This analysis should be based on a quantum-chemical description of this interaction. We are currently developing an appropriate theoretical model. However, we can assume that the amino-acids are captured between the unwound polymer fibers, thus forming an additional dense shell, which blocks an access of the amino-acids to the membrane surface, and therefore the quenching stops. Indeed, we used the geometry of the laser radiation grazing incidence on the membrane surface, whereas this additional shell is apparently formed at distance about 300 μm from the membrane surface (the exclusion zone size, see the monography [7] and our works [10,11]). Recall that in [20] it was shown that bacteria are effectively repelled from the Nafion surface precisely at this distance. This result proves once again that polymer fibers, unwound in the liquid bulk, actually perform the function of a glycocalyx, i.e., prevent the penetration of various foreign particles into the membrane. 

Based on comparisons of the results for ordinary water and water-based saline solution without amino-acids (Figure 2 and Figure 3) and for suspensions of amino-acids prepared in ordinary water-based saline solution, we can conclude that the additional shell formed by unwound polymer fibers and amino-acids, trapped between the fibers, prevent the access to the membrane surface not only for amino-acids, but also for water molecules, i.e., swelling slows down. 

The exception is for the glutamic acid suspension: its *I*(*t*) dependence is approximately the same for ordinary water-based and DDW-based NaCl solution, see Figure 6f. Apparently, glutamic acid is capable of penetrating through the unwound polymer fibers, i.e., for this amino-acid the luminescence quenching effect is the same both in an ordinary water-based suspension and in a DDW-based suspension. This should be considered as a specific feature of that amino-acid.

As shown in [11], luminescence centers are sulfonate groups, which, according to [1], are negative, and experience Coulomb attraction to positively charged amino-acids (these are lysine, and also, apparently, histidine and alanine, see [27] and pI data in Table 1). In addition, there is an attractive Coulomb interaction between negative sulfonate groups and neutral amino-acids (glutamic and aspartic acids, which are close to isoelectric point at pH = 3, see [27] and Table 1). To describe the interaction between amino-acids and the membrane surface, it is necessary to involve theoretical models based on dispersion interactions between charged particles and surfaces, including biological membranes, see [35]. Currently, we are developing appropriate theoretical approaches to describe the interaction between amino-acids and charged Nafion surface with taking into account the effect of unwinding.

As is known (see, for example, [33] and also [36,37,38,39]), luminescence quenching is controlled by charge transfer processes between the luminescence center and a quencher (in our case, an amino-acid). Since we are dealing with interaction of positive/neutral amino-acid with negative sulfonate group, the nature of luminescence quenching in our case is also due to charge transferring. Recall that a logical binary cell, based on the effect of luminescence quenching with the participation of aspartic acid, was implemented, see ref. [21]. As indicated in the Introduction section, one of the goals of our work was to study the possibility of luminescence quenching effects to create a binary cell in the “Nafion membrane-amino acid suspension” system taking into account the unwinding effect of polymer fibers/in the absence of this effect.

In this experiment, the Nafion plate was first soaked in glutamic acid suspension (1 g/L) prepared in an ordinary water-based saline solution of volume *V*; in this case, the luminescence signal reached zero level rather quickly, see Figure 6f. After 80 min of Nafion soaking, the same volume *V* of the lysine suspension (1 g/L, also prepared in ordinary water-based saline solution) was added. Dependences *I*(*t*) vs. the soaking time are presented in Figure 7. We can see that the addition of lysine suspension results in an abrupt increase in *I*(*t*) followed by a slow decay. The reason for the abrupt increase in *I*(*t*), in our opinion, is as follows: since at pH = 3 lysine is positive, while glutamic acid is neutral (see Table 1), they experience an attractive interaction, i.e., form a Coulomb complex on the membrane surface. In this case, we can talk about a change in the processes of charge transfer from the luminescence center (sulfonate group) to the luminescence quencher (glutamic acid), which can lead to the quenching termination. Recall that the purpose of this experiment was to study the possibility of creating a binary cell based on this system. Obviously, for suspensions of amino-acids prepared in physiological solution based on ordinary water, the implementation of a logical cell is impossible due to the nonstationary behavior of the luminescence intensity *I*(*t*) after recovering with lysine.

The case for DDW-based suspensions is presented in Figure 8a,b. Panel (a) corresponds to the adding a lysine suspension with concentration of 0.1 g/L (50-th minute), and panel (b) is related to the adding a lysine suspension with concentration of 1 g/L (80-th minute). The volume *V* of the lysine suspension was equal to the original volume *V* of the glutamic acid suspension for both cases. We can see that the lysine adding leads to an abrupt increase and reaching a quasi-stationary level of *I*(*t*). As follows from the graphs, this level is controlled by the lysine concentration: for lysine concentration of 1 g/L, the jump is much higher. The dependences for DDW can be conventionally regarded as a switching (triggering) effect during luminescence quenching. It is important that in the case of DDW the level of luminescence after recovering (adding lysine suspension) is approximate the same unlike the situation with ordinary water, see Figure 7. Note that triggering effects are used to create logical gates for computers. Thus, DDW-based amino-acids suspensions can be a promising media for developing computers based on biological processes.

If we add once again the volume *V* of a glutamic acid suspension (1 g/L), prepared in DDW-based solution of NaCl, to the mixture of glutamic acid-lysine suspensions (see Figure 8b), this will once again cause a sharp decrease in the luminescence signal *I*(*t*), see Figure 9. Thus, the alternate addition and removal of the lysine/glutamic acid suspension, prepared in DDW-based saline solutions, may be perspective for creating the logical binary cell 0–1–0 based upon biological particles. Note in this connection, that the creation of electronic devices based on biological objects is a dynamically developing topic in modern biology, see monograph [40]. We can imagine that amino-acids suspensions, causing the effects of quenching/restoring luminescence from Nafion, can be used for developing logical gates in biological computers.

## 5. Conclusions

1. It is important to take into account the effects of quenching luminescence in experiments with photoluminescence spectroscopy, where the interaction of amino-acids with Nafion membrane is studied.

2. The dynamic of Nafion membrane swelling dynamics and luminescence quenching by amino-acids are different for physiological solutions prepared on ordinary water (the deuterium content is 157 ppm) and DDW (the deuterium content is ≤ 1 ppm). The stationary luminescence intensity level *I*(*t*) for ordinary water is much higher than for DDW, which is due to the protective function of polymer fibers, unwound into the liquid bulk and trapped amino-acids.

3. The addition of a lysine suspension leads to an abrupt increase in the luminescence of Nafion subjected to prolonged soaking in a glutamic acid suspension. The magnitude of the jump depends on the lysine concentration and the deuterium content in the suspension.

4. The effects of quenching/recovering the luminescence of sulfonate groups due to the addition of amino acids with opposite charges can be used to create logical binary cell based on Nafion in amino-acid suspensions based upon DDW.

## Figures and Tables

**Figure 1 polymers-13-02635-f001:**
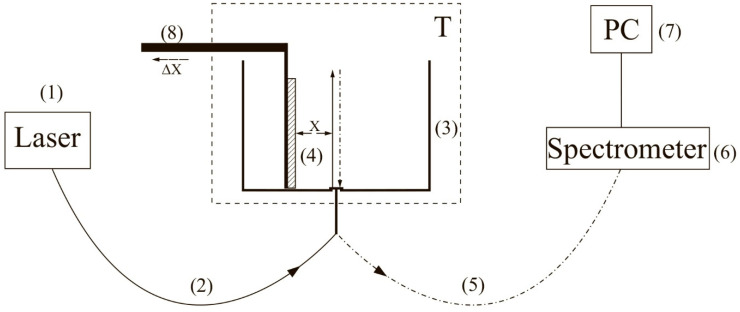
Schematic of the experimental setup. (1) laser, (2), (5) multi-mode optical fiber, (3) cell with a liquid sample, (4) Nafion plate, (6) spectrometer, (7) computer, (8) stage equipped with a micrometric horizontal feed for Nafion plate alignment relative to the optical axis, (T) thermostat.

**Figure 2 polymers-13-02635-f002:**
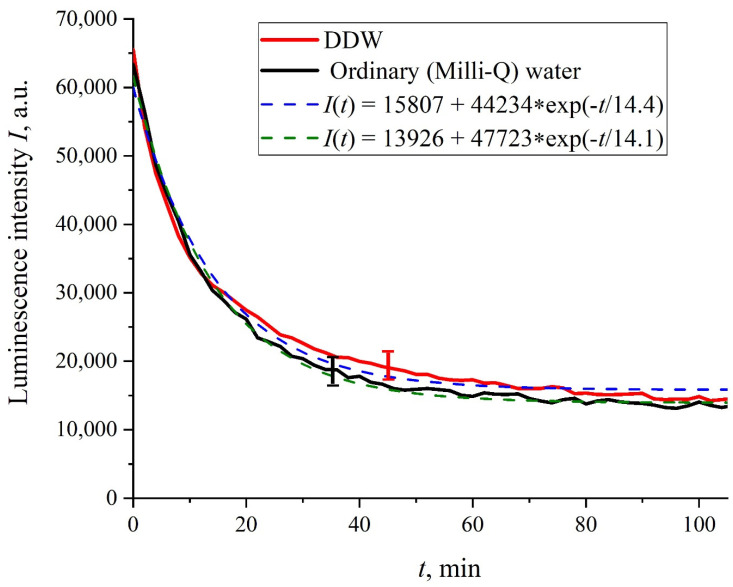
Intensity of luminescence *I*(*t*) vs. the soaking time in ordinary water and DDW in the absence of NaCl and amino-acids.

**Figure 3 polymers-13-02635-f003:**
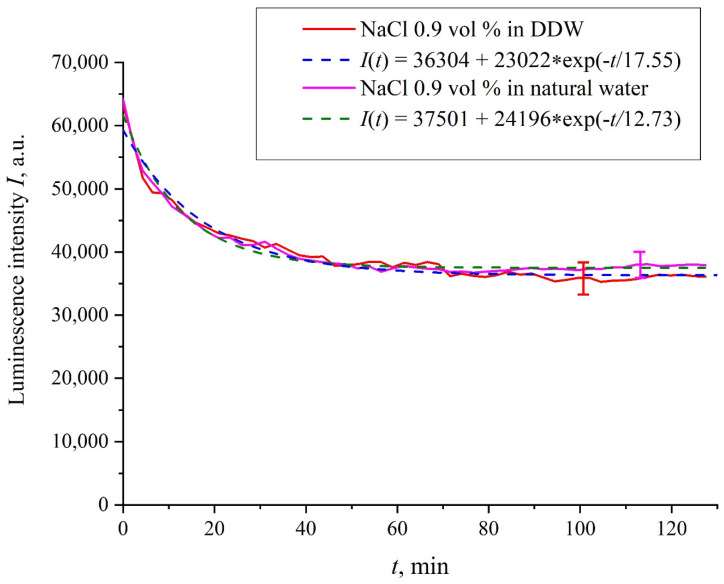
Intensity of luminescence *I*(*t*) vs. the soaking time in ordinary water-based and DDW-based physiological solutions (0.9 vol.% NaCl) without amino-acids.

**Figure 4 polymers-13-02635-f004:**
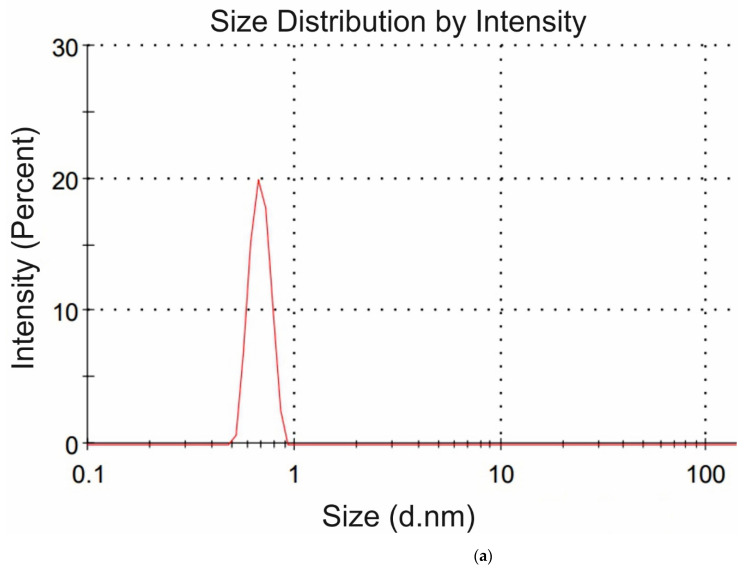
DLS intensity distribution over the particle diameters in suspensions of amino-acids at the scattering angle of 173°: (**a**) Lysine, (**b**) Glutamic acid.

**Figure 5 polymers-13-02635-f005:**
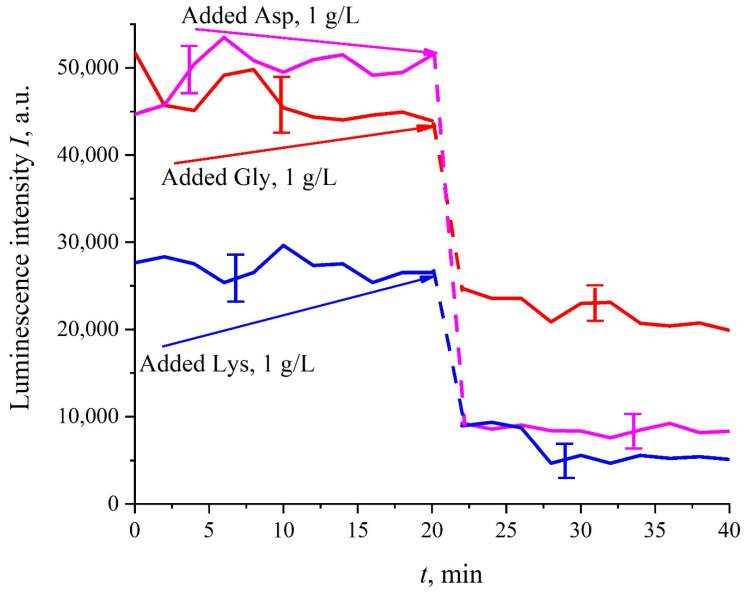
Dependence of the luminescence intensity from the colloidal mixture of Nafion particles at adding amino-acid suspensions (20-th minute). The dependences for aspartic acid, glycine and lysine with a concentration of 1 g/L are given.

**Figure 6 polymers-13-02635-f006:**
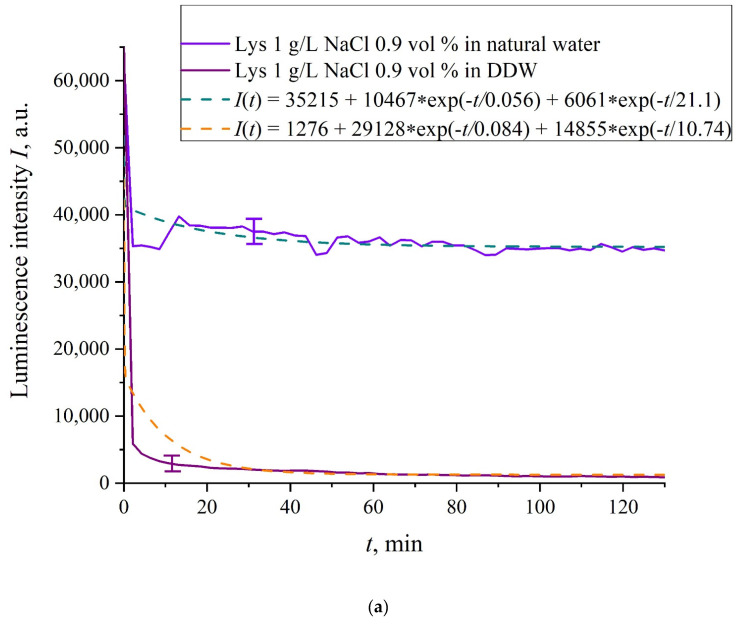
Luminescence intensity *I*(*t*) vs. soaking time in ordinary water-based and DDW-based saline solutions (0.9 vol.% NaCl) for the following amino-acid suspensions. (**a**) Lysine; (**b**) Histidine; (**c**) Glycine; (**d**) Alanine; (**e**) Aspartic acid; (**f**) Glutamic acid.

**Figure 7 polymers-13-02635-f007:**
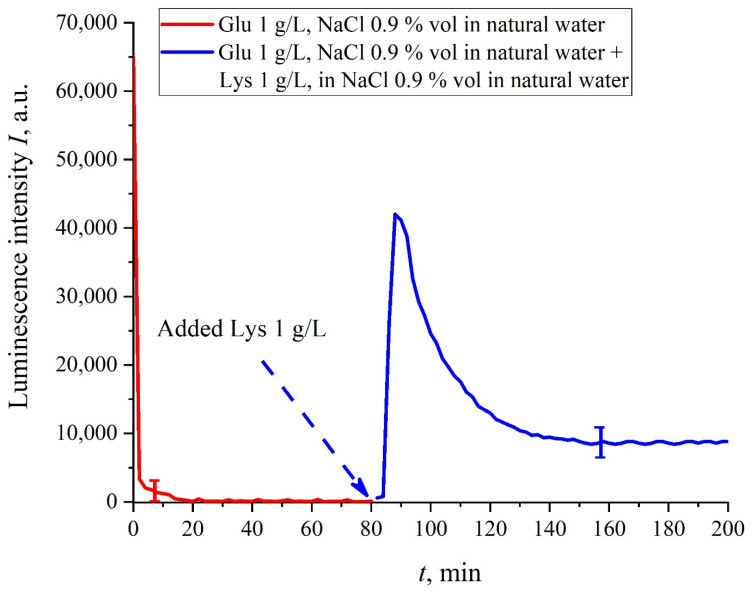
Dependence *I*(*t*) vs. the soaking time in an ordinary water-based saline solution (0.9 vol.% NaCl) at adding a lysine suspension (1 g/L), prepared in an ordinary water-based saline solution. The adding of lysine suspension occurred at 80-th minute. The volumes *V* of suspensions of glutamic acid and lysine are the same.

**Figure 8 polymers-13-02635-f008:**
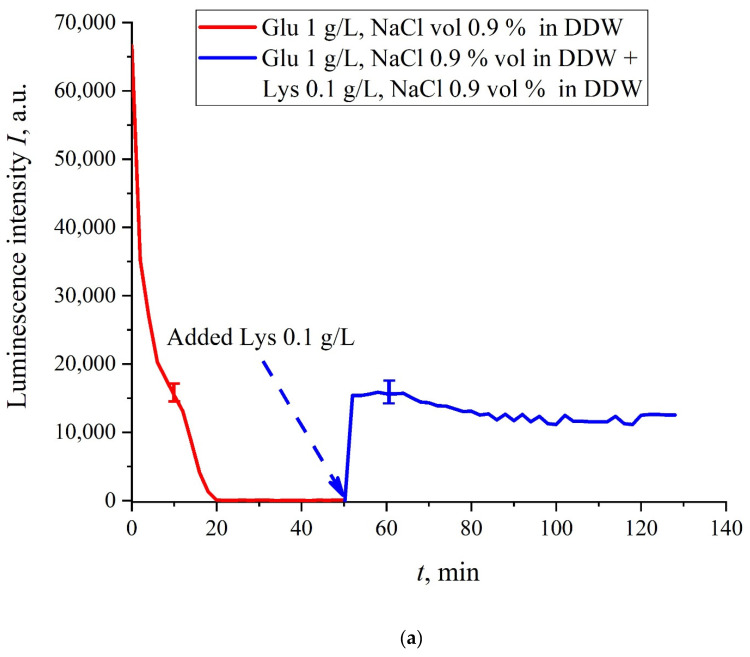
Luminescence intensity *I*(*t*) vs. the soaking time in DDW- based saline (0.9 vol.% NaCl) containing suspension of glutamic acid (1 g/L). A lysine suspension (1 g/L, DDW- based saline also) was added at 50/80-th minute; the volumes *V* of suspensions of glutamic acid and lysine are equal. Panel (**a**) corresponds to adding the lysine suspension with concentration 0.1 g/L at 50-th minute. Panel (**b**) corresponds to adding the lysine suspension with a concentration of 1 g/L at 80-th minute.

**Figure 9 polymers-13-02635-f009:**
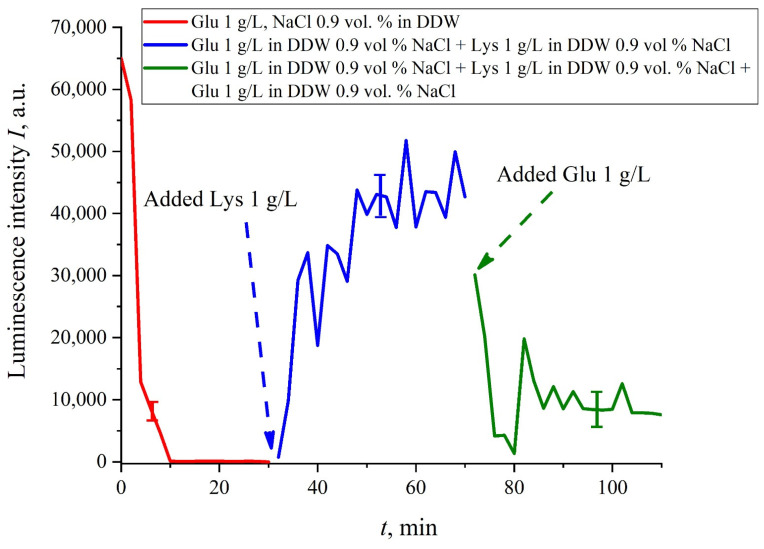
Luminescence intensity *I*(*t*) vs. soaking time in DDW- based salines (0.9 vol.% NaCl) containing the glutamic acid suspension (concentration 1 g/L). At the 30-th minute, a lysine suspension (1 g/L), prepared in DDW-based saline solution, was added; the volumes *V* of glutamic acid and lysine suspensions are the same. Then, at the 70-th minute, a glutamic acid suspension of the same volume *V* and concentration of 1 g/L, was added once again.

**Table 1 polymers-13-02635-t001:** Diameters, charges and pI values of amino-acids under study.

	Diameter (Å), Literature Data	Diameter (Å), Measured Data	Charge (10^20^ Q)	pI [27]
Lysine	5.7 [28]; 8.02 [29]	6.87 ± 1.0	1.59	9.8
Histidine	5.5 [30]; 8.9 [29]	4.52 ± 1.0	−0.837	7.6
Glycine	4.2 [30]; 5.1 [29]	-	-	-
Alanine	4.7 [31]; 4.6 [29]	6.78 ± 1.0	−0.063	6.01
Aspartic acid	5.0 [30]; 7.44 [29]	7.1 ± 1.0	−0.55	3.0
Glutamic acid	5.3 [30]; 7.98 [29]	5.98 ± 3.0	−0.52	3.2

**Table 2 polymers-13-02635-t002:** The rates *v*_1_ and *v*_2_ of decay of the luminescence intensity and the values of free constants *A* for suspensions of amino acids in ordinary water.

	*v*_1_, a.u.	*v*_2_, a.u.	*A*, a.u.
Lysine	10,467/(0.056 × 2.78) = 678,186	6061/(21.1 × 2.78) = 103	35,215
Histidine	5207/(0.077 × 2.78) = 24,307	21,844/(5.9 × 2.78) = 1330	38,057
Glycine	49,845/(1.1 × 2.78) = 16,288	10,226/(131 × 2.78) = 28	5612
Alanine	33,604/(8.3 × 2.78) = 1455	16,327/(111 × 2.78) = 53	12,032
Aspartic acid	26,024/(13 × 2.78) = 725	2010/(88.2 × 2.78) = 8.2	18,755
Glutamic acid	60,798/(1.81 × 2.78) = 437,081	2174/(1.81 × 2.78) = 432	778

**Table 3 polymers-13-02635-t003:** The rates *v*_1_ and *v*_2_ of decay of the luminescence intensity and the values of free constants *A* for suspensions of amino acids in DDW.

	*v*_1_, a.u.	*v*_2_, a.u.	*A*, a.u.
Lysine	29,128/(0.084 × 2.78) = 124,644	14,855/(10.7 × 2.78)= 497	1276
Histidine	5207/(0.035 × 2.78) = 125,398	52,298/(29.8 × 2.78) = 631	1521
Glycine	39,854/(1.6 × 2.78) = 8953	24,696/(43.4 × 2.78) = 205	415
Alanine	36,069/(0.07 × 2.78) = 185,216	10,412/(13 × 2.78) = 288	83
Aspartic acid	29,348/(0.03 × 2.78) = 351,642	15,459/(24.9 × 2.78) = 223	1539
Glutamic acid	31,700/(0.05 × 2.78) = 227,894	13,828/(7.54 × 2.78) = 659	1543

## Data Availability

The data presented in this study are available on request from the corresponding author.

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
