# Peer review of "Dynamics of Polymer Membrane Swelling in Aqueous Suspension of Amino-Acids with Different Isotopic Composition; Photoluminescence Spectroscopy Experiments"

_polymers, 2021, doi:10.3390/polym13162635_

Round 1
Reviewer 1 Report
In my opinion the present manuscript should be published in a more specialized journal. Conclusion 4 is far from what we can state from the results of this research.
Author Response
We are grateful to the reviewer for careful manuscript reading and very valuable comments. The referee's comments / questions are marked by Italic font, while our responds are in normal font.
In my opinion the present manuscript should be published in a more specialized journal. Conclusion 4 is far from what we can state from the results of this research.
We disagree with the reviewer opinion that the manuscript should be published in a more specialized journal. In the section Aims and scope of the Polymers journal (https://www.mdpi.com/journal/polymers/about) the Publisher had declared: “Polymers provides an interdisciplinary forum for publishing papers which advance the fields of… (iii) understanding of new physical phenomena. The articles topics in the Polymer Applications section (https://www.mdpi.com/journal/polymers/sections/Polymer_Applications) include, in particular, all kinds of applications (…from biomedical engineering to space engineering). In our opinion, our paper relates, among other things, to polymers for biomedical engineering purposes: a model of a biological trigger based on quenching/restoring the luminescence from Nafion surface sulfonate groups by alternately adding positively charged / neutral amino acids to a colloidal isotonic solution is proposed. Seemingly, we have made the significant progress in understanding new physical mechanisms of the polymer matrix interacting with aqueous suspensions amino-acids, having different isotopic compositions. Indeed, the dependence of the Nafion membrane surface luminescence intensity on the swelling time in water/saline solution with different isotopic compositions is described by approximately the same exponential functions, while in these saline solutions with amino-acids suspensions, the Nafion membrane surface luminescence intensity vs. swelling time significantly depends on the isotopic composition.
Conclusion 4 is far from what we can state from the results of this research.
We have completely rewritten the Introduction section. The need to rewrite the Introduction section is due to the fact that after submitting the manuscript to the journal Polymers, we found an article by G. Ji, T. Zheng, X. Gao, Z. Liu, "A highly selective turn on luminescent logic gates probe based on postsynthetic MOF for aspartic acid detection ", Sensors & Actuators: B. Chemical 284 (2019) 91–95, https://doi.org/10.1016/j.snb.2018.12.114, which appears to be the first to report the creation of a logical binary cell based on a system that includes an organometallic crystal matrix, into the cages of which trivalent terbium ions, divalent copper ions and aspartic acid are introduced. In this case, trivalent terbium ions are a very effective fluorophore, while bivalent copper ions serve as a luminescence quencher for this fluorophore. At the same time, aspartic acid forms a complex with copper ion, which neutralizes the luminescence quenching function of this ion, i.e. the luminescence of Tb3+ is efficiently recovered. This work shows that a high luminescence level (for example, in the absence of copper ions, or in the presence of both copper ion and aspartic acid) can be conventionally considered as "1" of a binary logic cell, while a low luminescence level in the presence of copper ions can be conventionally considered as "0" of the binary cell. In our opinion, our results are quite similar to those obtained in the mentioned work. Namely, in our case the effects of quenching / recovery of luminescence are observed in the "Sulfonate group - glutamic acid - lysine" system, while in the cited article the same effects are observed in the "Tb3+ - Cu2+ - aspartic acid" system. In other words, the results obtained in the mentioned work and our results can be considered as a fundamental multidisciplinary step to develop biological computing, which exercise the same principle of the charge transfer like ordinary silicon based transistor computers. Note that, in our case, it makes sense to make a logical cell based on DDW, since when using suspensions of amino acids based on ordinary water, the level of reduced luminescence decreases (see Fig. 7), while in the case of suspensions based on DDW the level of the recovered luminescence remains approximately constant, see Fig. 8.
In this regard, we decided to formulate conclusion 4 of our manuscript as follows: The effects of quenching / recovering the luminescence of sulfonate groups due to the addition of amino acids with opposite charges can be used to create logical binary cell based on Nafion in amino-acid suspensions based upon DDW.
Reviewer 2 Report
the manuscript is presenting an interesting topic on PL experiments performed on Aqueous Suspension of Amino-acids With Different Isotopic Composition; the few comments and questions are due before we proceed with this submission:
-there are some literature which contradict glutamic acids being negatively charged,how this related to ion concentration. please explain.
-citation is required to: 10.1016/j.spmi.2020.106677
-fig. 6c shows a slightly observable difference between the model and the data. how this is justified?
-is there any reasons to explain the physical nature of the relaxation process in fig 6b?
when authors take my comments in and answer my comments I can reconsider my decision.
Author Response
We are grateful to the reviewer for careful manuscript reading and very valuable comments. Below we provide detailed answers to the reviewer questions/comments. The referee's comments / questions are marked by Italic font, while our responds are in normal font.
there are some literature which contradict glutamic acids being negatively charged, how this related to ion concentration. please explain.
Information about the charge sign of amino acids was taken from the site https://chem.libretexts.org/Bookshelves/Organic_Chemistry/Map%3A_Organic_Chemistry_(Wade)/25%3A_Amino_Acids_Peptides_and_Proteins/25.02%3A_Isoelectric_Points_and_Electrophoresis
According to these data, the isoelectric point for Glutamic acid is pI = 3.2, i.e. when pH = 3 (these values ​​ are attained near the Nafion – water interface), Glutamic acid should possess a positive, rather than a negative charge. In accordance with the above hyperlink, in order to correctly measure the amino-acid charge sign, a solid matrix layer consisting of paper or a cross-linked gelatin-like substrate were chosen and impregnated with an ionic buffer. A small amount of an amino-acid, peptide or protein sample is then placed near the matrix strip center and an electrical potential is applied to the strip tips. In this case, the matrix solid structure prevents the diffusion of solute molecules, which will remain in the previous location, unless an electrostatic potential is applied. It is obvious that in this case an amino-acid net charge value could be measured, i.e. +e, or –e, or 0. In our research, we are dealing with amino-acids in an aqueous salt solution, i.e. charged amino-acids are surrounded by a shielding cloud of counter-ions. In this case, the charge absolute value will always be less than the charge of the electron; this is exactly what was obtained in our Zetasizer- assisted experiments. According to the above hyperlink, the pI value for Lysine is 9.8, i.e. at pH = 3, Lysine will have a positive charge even with counter-ions shielding. Thus, in our opinion, it makes no sense to compare the literature data on the amino acid charge sign with our experimental results. All these considerations are outlined in the new version of the manuscript; undoubtedly, the corresponding section in the new version looks more convincing.
citation is required to: 10.1016/j.spmi.2020.106677
In the new version, we are cited this work, see Ref. [2].
fig. 6c shows a slightly observable difference between the model and the data. how this is justified?
Unfortunately, we did not quite understand this question. We do not incorporate any models into our considerations. In this figure, we illustrate different dependences for a Glycine suspension prepared in ordinary water-based and DDW-based physiological saline. It is seen that the experimental data are approximated by different exponential functions, and the differences, in our opinion, are quite noticeable. Perhaps, the reviewer meant the dependencies in Fig. 6 (f) for a Glutamic acid suspension prepared in ordinary water-based and DDW-based physiological saline. In this case really, there is practically no difference between the dependence of the Nafion surface luminescence intensity on the swelling time, despite the fact that for DDW, the effect of polymer fibers unwinding from the polymer surface being negligible, whereas for ordinary water this effect manifests clearly, see. Fig. 6 (a) - (f). Our hypothesis is that the luminescence quenching effect (a sharp drop of the luminescence intensity) occurs due to the contact of a negatively charged sulfonate group and an amino-acid; thus, the charge transfer from the sulfonate group to the amino-acid is plausible, which leads to the quenching effect. According to our approach, when polymer fibers unwind into the liquid bulk, a rigid "brush" of polymer fibers is formed; the length of this brush is about 300 microns from the membrane surface. Amino-acids are located in between the polymer fibers, which brings extra density to this "brush". This is why, new amino-acids no longer have access to the membrane surface, and since the membrane surface luminescence is studied in our experiments, it should be expected that for ordinary water-based amino-acid suspensions, the quenching effect should disappear. Apparently, Glutamic acids are capable of penetrating through the unwound polymer fibers, i.e. the luminescence quenching effect will be the same both in an ordinary water-based suspension and in a DDW-based suspension. This should be considered as a specific feature of that amino-acid. In the new version, we discuss this issue in more detail.
is there any reasons to explain the physical nature of the relaxation process in fig 6b?
In this case, the luminescence dynamics is described by two exponentials, which corresponds to two dynamic modes. The first one is very fast and related to the quenching. The second one is related to some very slow relaxation process. If a Histidine suspension is prepared in ordinary water-based physiological solution, then the first exponential is 5207×exp(-t/0.077), the second exponential is 21844×exp(-t/5.9), and a free parameter (luminescence intensity at t → ∞) is 38057 a.u. At the same time, for a suspension prepared in DDW-based saline, the first exponential is 12210×exp(-t/0.035), the second exponential is 52298×exp(-t/29.8), and the free parameter is 1521 a.u. For an ordinary water-based Histidine suspension, the first exponent decrease rate by a factor of e = 2.781828 (the Euler number) is v1 = 5207/(0.077×2.781828) = 24307 a.u., and for a DDW-based suspension, we obtain v1 = 12210/(0.035×2.781828) = 125398 a.u. However, it should be noted that these estimates are very approximate, since our experimental setup is not designed to measure such high luminescence quenching rates. In the case of a ordinary water-based suspension, the second exponent decrease rate for Histidine is diminished by Euler number e = 2.781828 and the corresponding velocity is v2 = 21844/(5.9×2.781828) = 1331 a. u., while for a DDW-based suspension we obtain v2 = 52298/(29.8×2.781828) = 631 a. u. Thus, for both suspensions, the second exponents decrease approximately with the same rate, but in the case of an ordinary water-based suspension, the free parameter is 25 times larger than the free parameter for the DDW-based suspension. We can say that in the case of an ordinary water-based histidine suspension, the luminescence quenching interrupts due to the polymer fibers unwinding, and for a DDW-based suspension, the luminescence quenching is completed. Thus, our approach is applicable to a Histidine suspension, albeit with certain limitations. In the new version, we discuss this issue in more detail. In addition, in the new version in Tables 2 and 3, instead of the values of τ1 and τ2, we present precisely the luminescence intensity decrease rate divided by e = 2.781828 for the first and second exponentials. We show in the new version that the range of the v1 / v2 velocity ratio is approximately the same for amino-acids based on ordinary water and DDW, i.e. decay rates v1 and v2 are not the key parameter characterizing the differences between amino acids in ordinary water and DDW. We show that the key parameter is the difference between free constants. We discuss this in the new version of the manuscript. In our opinion, this improves the presentation quality.
Reviewer 3 Report
The manuscript is presenting an interesting work of photoluminescence spectroscopy experiments where the interaction mode of the polymer membrane made from Nafion with various amino acids was studied.
The work is well and logically organized. The introduction section is correct and suitable references are presented. However, the cited works are quite old. Please try to find more up-to-date references. Even though it is research work the literature survey needs to be done and to highlight the problem based on the recent papers. Make sure that all required information in the reference list are presented.
Due to the fact that the idea of the work is really interesting, did the authors consider studying other polymers? In the presented version the study has been limited to the Nafion membranes only.
Please be aware that Nafion is a trade mart so please use the suitable labeling, i.e. ™.
Line 207 – “acids; It can” it should be “acids. It can”
Did the authors consider to made the material characterization of the polymeric materials before and after measurements? Did you evaluate that the developed method can influence/destabilize the membrane etc.?
Fig. 4 please provide better quality results. The best to draw the plots by the authors based on the collected data.
Table 1 – the column presenting diameter – it is not clear that the presented is a range of the diameter? For instance for Lysine is it a range from 5.7 to 8.02 ? Please present in the clear way. Moreover, present the data with the same accuracy.
Fig. 5 – How the time was selected? Is it not too short 40min only?
Fig. 6C – use the uniform way of presentation, in all other plots there are abbreviations used. So use also here.
Table 2 and 3 the captions are missing.
The work is interesting but in many points the scientific explanation of the results are missing. I would suggest to do so and present much deeper explanation of the established data.
Author Response
We are grateful to the reviewer for careful manuscript reading and very valuable comments. Below we provide detailed answers to the reviewer questions / comments. The referee's comments / questions are marked by Italic font, while our responds are in normal font.
The work is well and logically organized. The introduction section is correct and suitable references are presented. However, the cited works are quite old. Please try to find more up-to-date references. Even though it is research work the literature survey needs to be done and to highlight the problem based on the recent papers. Make sure that all required information in the reference list are presented.
We are grateful to the reviewer for this comment. The Introduction has been completely rewritten and more relevant references have been introduced.
Due to the fact that the idea of the work is really interesting, did the authors consider studying other polymers? In the presented version the study has been limited to the Nafion membranes only.
We are grateful to the reviewer for this question. In the future, we are going to study anion-exchange membranes (recall that Nafion is a proton-exchange membrane). Anion exchange membranes are designed to transmit anions, for instance, hydroxide anions. We expect a number of interesting new results with this membrane.
Please be aware that Nafion is a trade mark so please use the suitable labeling, i.e. ™.
We agree with the reviewer. In the very first line of the manuscript new version, in the Introduction section, we note that Nafion is precisely a trademark.
Line 207 – “acids; It can” it should be “acids. It can”
This annoying typo has been corrected. We are grateful to the reviewer for this remark.
Did the authors consider to make the material characterization of the polymeric materials before and after measurements? Did you evaluate that the developed method can influence/destabilize the membrane etc.?
This is a very good question. Of course, the membrane oxidizes during long-term measurements. At the same time, optical radiation irradiates the membrane in the grazing incidence geometry, with the pump wavelength λ = 369 nm, which corresponds to the long-wavelength boundary of the Nafion absorption band. Therefore, heating and membrane degradation during irradiation does not occur. At the same time, when we operated pumping radiation near the Nafion spectral absorption maximum (λ = 270 nm), then in this case the luminescence degrades quickly enough, which is associated with heating. For more detail see our recent work Bunkin, N.F.; Shkirin, A.V.; Kozlov, V.A.; Ninham, B. W.; Uspenskaya, E.V.; Gudkov, S.V. Near-surface structure of Nafion in deuterated water. J. Chem. Phys. 2018, 149, 164901, ref. [11] in the new version of the manuscript.
Fig. 4 please provide better quality results. The best to draw the plots by the authors based on the collected data.
Thanks for this remark. The quality of this drawing has been significantly improved in the new version of the manuscript.
Table 1 – the column presenting diameter – it is not clear that the presented is a range of the diameter? For instance or Lysine is it a range from 5.7 to 8.02? Please present in the clear way. Moreover, present the data with the same accuracy.
Table 1 (second column) presents the hydrodynamic diameters of amino-acids (double radii) in the dynamic light scattering experiment. The first column presents the literature data with appropriate references. We consider as a great piece of luck to detect several Angstroms-sized scatterers in our dynamic light scattering experiment with laser wavelength l = 0.63 μm. The literature data given in the Table 1 show that we, most likely, were not very much erroneous in measuring the sizes of amino-acids. The relevant comments are added to the new version of the manuscript.
Fig. 5 – How the time was selected? Is it not too short 40 min only?
We have chosen the experiment duration of 40 minutes just for the time saving. As follows from the graphs in this Figure, before adding amino-acid suspensions to the micron-sized Nafion particles colloidal solution, the luminescence level was approximately the same and was determined by the Nafion particles concentration (to improve the quality of the presentation, we took solutions with slightly different concentrations of Nafion particles). By the 20-th minute, amino acid suspensions were added to the colloidal solutions. As is known, luminescence quenching is a very fast process, see, for example, Lakowicz, J.R. Principles of Fluorescence Spectroscopy, Chapter 8, Quenching of Fluorescence, pp. 277-330, Springer, Boston, MA, 2006, ref. [37]. In our case (see Fig. 5) quenching occurs within 1 - 2 minutes, which is, obviously, due to the processes of mixing amino-acid suspension inside the Nafion particles solution. As follows from the dependencies shown in Fig. 5, after adding the amino-acid suspension and the completion of the hydrodynamic processes of mixing, the luminescence level again reaches a stationary level. Thus, the duration of this experiment is not crucial.
Fig. 6C – use the uniform way of presentation, in all other plots there are abbreviations used. So use also here.
Thanks for this comment. Fig. 6C was corrected in accordance with the reviewer recommendation. The same was done for Fig. 5, Fig. 6 f, Fig. 7, Fig. 8, Fig. 9, and Graphial Abstract.
Table 2 and 3 the captions are missing.
This has been fixed in the new version. Captions have been inserted for all three tables.
The work is interesting but in many points the scientific explanations of the results are missing. I would suggest to do so and present much deeper explanation of the established data.
The text of the manuscript was completely rewritten. Note that this work is purely experimental. Unfortunately, we do not have a quantitative mathematical model describing the discovered phenomena, and therefore the description presented is completely qualitative. Currently, we are developing quantum-chemical models, on the basis of which it should be possible to obtain a quantitative description of the described effects, in particular, luminescence quenching due to charge transfer from the luminescence center (sulfonate group) to an amino acid, as well as the triggering effect (switching the processes of quenching/recovering luminescence upon mixing amino-acid suspensions of opposite electric charges). As soon as the theoretical results are approved at scientific seminars, we will prepare a new manuscript.
Round 2
Reviewer 1 Report
Thank for your answers to my comments.
Reviewer 3 Report
The authors took into account all of the comments and suggestions. The manuscript can be published in the current version.